# The Role of Midwives in the Course of Natural Childbirth—Analysis of Sociodemographic and Psychosocial Factors—A Cross-Sectional Study

**DOI:** 10.3390/ijerph192315824

**Published:** 2022-11-28

**Authors:** Patrycja Guzewicz, Matylda Sierakowska

**Affiliations:** Department of Integrated Medical Care, Medical University of Bialystok, ul. M. Skłodowskiej—Curie 7A, 15-096 Białystok, Poland

**Keywords:** labor pain, non-pharmacological labor pain-relief methods, midwife, childbirth

## Abstract

Background: An important role in the course of natural childbirth is played by midwives, who should effectively work on relieving pain. This study aims to present the opinions of midwives on non-pharmacological methods of relieving labor pain; the frequency of their use and reasons for their abandonment; and the relationship between the use of non-pharmacological methods of relieving labor pain and perceived job satisfaction, burnout, and self-efficacy of the midwife. Methods: The study was conducted online, with the participation of 135 Polish midwives working in the delivery room. The author’s survey questionnaire, the Generalized Self-Efficacy Scale (GSES), the LBQ Burnout Questionnaire, and the Scale of Job Satisfaction were used. Results: Among the surveyed midwives, 77% use vertical positions in work with a patient giving birth. Almost all respondents consider vertical positions as an example of a non-pharmacological method of relieving labor pain; those with master’s degree felt more prepared for their use (*p* = 0.02). The most common reason for abandoning their use was disagreement on the part of co-workers (*p* = 0.005). An association was observed between the use of vertical positions and the level of burnout (*p* = 0.001) and a significant correlation between preparation for their use and self-efficacy assessment, burnout, and job satisfaction. Conclusion: Our research shows that it would be important to conduct additional training on the use of non-pharmacological methods to relieve labor pain and to present their benefits. In contrast to other research results, our results showed that midwives feel well prepared to use these methods; however, similar to other research, we found that they often feel disagreement from colleagues and a lack of support from their leaders. The use of vertical positions is related to burnout.

## 1. Introduction

Childbirth is one of the most important events in a woman’s life. Its course depends not only on the woman giving birth, but also on the midwife assisting in the childbirth. Her tasks include primarily medical duties, but also establishing an appropriate relationship with the woman giving birth and giving her support [1,2]. Patients often have false expectations of the childbirth, and they are often not prepared for labor contractions and also do not know how to deal with the pain. The midwife should provide the patient with information about the course of childbirth, suggest ways of coping with pain, and support the patient in active participation in childbirth. It is important for the midwife to develop a relationship with the patient in which they trust each other, because such a relationship can result in a positive experience of childbirth [3,4].

It is very important that the midwife constantly updates her professional knowledge and uses the acquired new skills in practice, in accordance with the principles of EBM (Evidence-Based Medicine). Nigerian studies showed that educating midwives and improving their qualifications has an impact on their practice and ability to cope with the pain of childbirth. It is therefore recommended that midwives constantly educate themselves and update their knowledge on the latest trends [5]. In order to relieve labor pain, vertical positions are increasingly used, as well as aromatherapy, music therapy, massage, and warm compresses [6,7]. These methods not only reduce the sensation of pain, but also affect the duration of the first and second periods of labor, help the mother to focus, and affect the type of injuries to the perineum during natural childbirth [8,9].

According to the recommendations of the WHO (World Health Organization), the patient should have a choice of the position in which she wants to give birth. Therefore, she does not have to give birth in a lithotomy position but should be encouraged to take vertical positions. These positions carry many advantages, including supporting the progression of childbirth by using the force of gravity, reducing pressure on the inferior vena cava, and helping with more effective contractions and better pelvic alignment [10,11]. A sense of strong anxiety causes the stimulation of the sympathetic nervous system and the release of adrenaline, cortisol, and norepinephrine. This results in increased labor pain and increased labor time [12].

Aromatherapy is one of the non-pharmacological methods of relieving labor pain. This technique reduces the sensation of labor pain and anxiety and also affects a lower rate of epidural anesthesia and opioid injections [13]. In aromatherapy, geranium oil, incense, lavender, rose, chamomile, jasmine, mint, orange, and clove oil are most often used [14].

In order to reduce the sense of anxiety and, thus, reduce the pain felt, music therapy is also used. Music stimulates the auditory center, which affects the limbic system, reticular formation, hypothalamus, and cerebral cortex. Because the auditory center and the pain center are located near each other, stimulation of the former can inhibit the pain center and the sensation of labor pain [15]. Studies have shown that it can have a beneficial effect on managing pain and anxiety during childbirth, mainly in women who give birth for the first time [16].

Massage given to a pregnant woman is a pain-relief technique that has been known for many years. It stimulates the production of endorphins and oxytocin, and it inhibits the secretion of adrenaline and norepinephrine. This method shortens the first and second periods of labor and also reduces the sensation of labor pain [17,18].

Heat therapy is one of many non-pharmacological methods of relieving labor pain. It has few side effects, and its use significantly affects childbirth. Warm compresses shorten the duration of labor, suppress the pain caused by labor contractions, and stimulate the contractile function of the uterus without negatively affecting the fetus. Heat therapy can, therefore, also be used to stimulate uterine contractions without the need to administer exogenous oxytocin. In addition, the use of warm compresses protects the perineum from rupture [19,20].

The aim of the study was to examine the level of knowledge and opinion of midwives about vertical positions and other non-pharmacological methods of relieving labor pain; identify the factors related to the abandonment of vertical positions and other non-pharmacological methods of pain relief by midwives; learn the opinions of midwives on the relationship between the organization of the work of the delivery room, the quality of the relationship established with the patient, the participation of the accompanying person during the course of natural childbirth; and determine the relationship between seniority and education, job satisfaction, burnout and self-efficacy, and the use of vertical positions and other non-pharmacological methods of relieving labor pain.

## 2. Materials and Methods

The research addressed midwives working in delivery rooms in healthcare facilities in Poland. The study group consisted of 135 midwives—134 women and 1 man.

The cross-sectional survey was conducted between February and April 2022, using an online survey created in the Google Form software. An invitation to participate in the study was sent to midwives via social media. All respondents were Poles.

Inclusion criteria: current work in the delivery room.

Respondents’ responses were recorded on the Google platform they were using. Participation in the study was voluntary. The studies were anonymous, and any participant could withdraw from the study at any time. Joining the study was tantamount to agreeing to participate in the study. The convenience sample method was used in order to obtain data.

The research used the diagnostic survey method, using the following research tools:Questionnaire designed by the study’s author (25 questions), containing imprint data (7 questions) and referring to the use of vertical positions and other non-pharmacological methods of relieving labor pain, factors related to the behavior of the woman giving birth during childbirth, self-assessment of the midwife’s preparation for the use of vertical positions, and non-pharmacological methods of relieving labor pain and difficulties in their use.Generalized Self-Efficacy Scale (GSES) [21];Link Burnout Questionnaire (LBQ) [22];Zalewska’s Job Satisfaction Scale (SPP) [23].

The research was carried out in accordance with the Declaration of Helsinki and Good Clinical Practice in research. The Bioethics Committee of the Medical University in Bialystok, Poland, granted the ethical approval for the study (APK.002.19.2022.). Participation was voluntary, and participants were informed about the project.

## 3. Results

### 3.1. General Characteristics of the Study Group

A total of 135 midwives participated in the survey. The most representative group (over 44%) was people aged 22–30. More than half of the midwives surveyed (60%) had a master’s degree.

The largest group of respondents was distinguished by short seniority, i.e., 0–5 years (35.6%), and the smallest group was midwives working 6–10 years in the profession (17.0%). Every second person completed a professional specialization. The examined midwives declared their own effectiveness (GSES), i.e., a general belief in the effectiveness in dealing with emerging difficulties and obstacles, on average at 30 points (s = 3.3), which indicates a high sense of self-efficacy.

The average level of burnout in the study group (LBQ), in the aspect of psychophysical exhaustion, was 18.1 points in the study group (s = 5.9), lack of commitment to relations with clients (patients) was 15.0 points (s = 4.7), sense of lack of professional effectiveness was 13.1 points (s = 4.0), and disappointment was 13.4 points (s = 5.7), which proves that the highest level of burnout occurs in the area of mental exhaustion, and the lowest in the category of disappointment and lack of professional effectiveness. After analyzing the overall assessment of job satisfaction (SPP), an average score of 4.7 points (s = 1.0) was obtained, which indicates a fairly high level of job satisfaction (Table 1).

### 3.2. Use of Vertical Positions and Other Non-Pharmacological Methods of Relieving Labor Pain

Among the surveyed midwives, as many as 77% declared that they use vertical positions in work with a patient giving natural childbirth. The most commonly used positions in practice are knee–elbow (69.6%), on all fours (44.4%), standing (43.7%), and sitting (40.7%). Virtually all respondents consider vertical positions as an example of a non-pharmacological method of relieving labor pain and confirm their beneficial effects on the course of natural childbirth (94.1%).

When analyzing other methods of relieving labor pain, half of the midwives admit that they do not use warm compresses (51.1%). About 3/4 of midwives use back massage as a method of pain relief for a pregnant woman (75.6%).

Aromatherapy is rarely used in the delivery room, every fourth midwife surveyed uses it often (15.6%) or sometimes (8.9%). Relieving labor pain with music is used by about half of the midwives surveyed (47.4%). The majority of respondents (85.2%) consider the methods of pain relief mentioned above to be effective.

Midwives in surveys also rated their degree of preparation for the use of vertical positions. On a scale from 1 to 5, almost 1/3 of midwives gave themselves 4 points (*I rather feel prepared*) (36.3%), and points 1–2 (*I do not have knowledge and skills—I rather do not feel prepared*) were given the least (1.5–11.1%).

The surveyed medical staff definitely appreciate the impact of the relationship between the woman giving birth and the midwife during the childbirth (97%). As the main reasons for not using pain-relief techniques at work, midwives mainly list disagreement on the part of colleagues (67.4%), management’s lack of support (58.5%), shortages in specialized equipment (48.1%), and insufficient knowledge of the midwife (45.2%).

In the opinion of midwives, women giving birth have rather average preparation for childbirth (58.5%). The factors that most strongly affect the behavior of a pregnant woman in the delivery room, according to the respondents are preparation of the pregnant woman (95.6%), previous experience of the midwife (83.7%), the presence of a loved one (79.3%), and cooperation with medical staff (76.3%). More than 70% of the surveyed midwives recognize the positive impact of the accompanying person on the course of childbirth (Table 2).

The midwives surveyed are of the opinion that in order to increase the use of non-pharmacological agents in the course of childbirth, more training on this subject should be organized for staff. A significant role would also be played by improving the equipment of hospitals with specialized equipment as well as increasing the number of staff in maternity wards (Figure 1).

### 3.3. Professional Status and the Use of Vertical Positions and Selected Non-Pharmacological Methods of Relieving Labor Pain

Midwives with a master’s degree were slightly more likely (83 vs. 68%) to use vertical techniques during childbirth. When it comes to the use of specific positions, there are some statistically significant differences from the education of midwives—those with higher education use the on all fours position more often (53 vs. 31%) (*p* < 0.05), and those with undergraduate education the position on a chair (*p* < 0.05).

The frequency of use of individual non-pharmacological methods of pain relief during natural childbirth does not depend on the education of the midwife (*p* > 0.05).

Another analysis concerns the self-assessment of the degree of preparation of the midwife for the use of vertical positions, depending on the level of education. Due to the ordered nature of the answer to this question, the Mann–Whitney test was used to assess the significance of the differences between the two groups. The *p*-test probability values are below 0.05, which means that midwives with a master’s degree rate their competence higher.

The level of preparation of women for childbirth is assessed similarly by both groups of midwives. On the other hand, the role of the accompanying person during childbirth is more appreciated by midwives with a master’s degree (*p* < 0.05) (Table 3).

In the same way, an attempt was made to determine the relationship of seniority with the opinions of midwives regarding the use of vertical positions, as well as non-pharmacological pain-relief measures during natural childbirth. Seniority does not in principle differentiate the frequency of use of vertical positions, although it can be noted that they are slightly more often used by midwives with an average seniority: 6–10 and 11–20 years.

On the other hand, the views of midwives of different seniority on the occurrence of most of the indicated difficulties in the use of pain-relief techniques are different (Table 4 and Table 5). Midwives with shorter tenure indicate more difficulties, which may be due to their actual occurrence, or may be due to less experience in overcoming these difficulties.

The shorter the length of service, the more often the midwives indicated disagreement on the part of co-workers (*p* = 0.02) and the head nurse or the head of department (*p* = 0.005), lack of equipment, materials, and employees (*p* = 0.02) and practical skills (*p* = 0.005).

A relationship was shown between the opinion on the role of the accompanying person in childbirth and seniority of the surveyed midwives. A negative sign of the correlation coefficient shows that midwives working shorter hours attach greater importance to the influence of the presence of the accompanying person on the course of childbirth (r_S_ = −0.19; *p* = 0.03).

Seniority differentiates the opinion of the midwives surveyed about two factors that could increase the use of non-pharmacological pain-relief agents. Midwives working longer pay slightly less attention to the organization of training (*p* = 0.02), and they less often indicate the need to improve the equipment of delivery rooms (*p* = 0.002).

### 3.4. The Use of Vertical Positions and Other Non-Pharmacological Methods of Relieving Labor Pain and Job Satisfaction, Self-Efficacy, and Burnout

There were no statistically significant relationships between self-efficacy and job satisfaction and the use of vertical techniques (*p* < 0.05). However, a statistically significant relationship between the level of burnout and the use of vertical positions was demonstrated. In people who do not use vertical positions, the values indicating burnout are highest in the area of professional effectiveness (*p* = 0.0001), disappointment (*p* = 0.0001), and lack of commitment to customer relations (*p* = 0.004).

Subsequent results present a correlation analysis between self-assessment of preparation for the use of vertical positions and assessment of work (effectiveness, burnout, and satisfaction). The analyses allowed us to observe that the use of vertical positions is an important element of the work of midwives, because the assessment of skills in this area is significantly correlated with most measures determining the general attitude to work (the exception is the measure of psychophysical exhaustion).

It has been shown that the better the preparation for the use of vertical positions, the higher the assessment of self-efficacy (r_S_ = 0.34; *p* = 0.0000) and job satisfaction (r_S_ = 0.26; *p* = 0.003). The greater the burnout—especially for measures of disappointment and lack of professional effectiveness—the lower the self-esteem of preparation for the use of vertical positions (r_S_ = −0.34 and −0.38; *p* = 0.0000).

Subsequent results provide an assessment of the link between the use of non-pharmacological methods for pain relief in terms of self-efficacy, burnout, and job satisfaction. There were no statistically significant relationships for most of the relationships considered, except for one of the forms of pain relief, namely back massage of pregnant women.

For this form of pain relief, there were negative correlations with the level of burnout (*p* = 0.00) (Table 6).

## 4. Discussion

The literature reports indicate that vertical positions serve not only to relieve labor pain, but also shorten the duration of the second period of childbirth, reduce the likelihood of obstetric complications, have a positive effect on the patient’s psycho-emotional state, and promote the progress of childbirth [24,25,26]. In addition to the benefits for the mother, it is also worth mentioning that vertical positions have a positive effect on the fetus by relieving the blood vessels [27].

The aim of the study was to present the opinions of midwives on non-pharmacological methods of relieving labor pain, the frequency of their use, and the reasons for abandonment, as well as the relationship between the use of non-pharmacological methods of relieving labor pain and perceived job satisfaction, burnout, and self-efficacy of the midwife of the delivery room.

Overall, it has been shown that only 3.7% of midwives do not use vertical positions in working with the patient, and only 0.7% of midwives believe that they do not serve to relieve labor pain. Given the benefits of using them, it is comforting that only such a small percentage of the surveyed midwives do not use, in their professional work, positions other than those involving lying down.

In China, the lithotomy position is the most commonly used position, and in France, 87.6% of midwives declared that they preferred patients to take a lying-on-their-back position [28,29].

This is probably due to the fact that vertical positions are less convenient for taking births. However, in studies conducted on Iranian patients, it was shown that, in the active second phase of the labor, the squatting position significantly reduces the severity of perceived labor pain when compared to the lithotomy and sitting positions [30].

American research has indicated that the positions most often proposed during the second period of childbirth are the squatting position and knee–elbow position, and less often sitting and on the side [31]. These results differ from the results of our research, in which the most frequently proposed position was knee–elbow, all fours, and standing.

The use of vertical positions and methods of pain relief may have its limitations, related not only to the knowledge and skills of the midwife, but also to factors resulting from the organization of the work of the delivery room [32]. In our research, midwives most often pointed out the following difficulties: the lack of cooperation with other medical staff; the lack of support from management; and the lack of equipment, materials, and a sufficient number of employees. The respondents also drew attention to the importance of training in this area. In research conducted in Saudi Arabia, it has been shown that the barriers to the use of non-pharmacological methods are lack of time, regulatory issues, lack of knowledge, patient unwillingness, and strong beliefs of epidural [33]. Moreover, nurses in Asmara indicated several barriers: lack of knowledge, inadequate time, lack of time, shortage of nurses, failure in administrative support, and resource scarcity [34]. Another research study has indicated that the main barriers to the use of non-pharmacological techniques are inadequate staffing, time constraints, limiting hospital policy, limited number of equipment, lack of resources, and lack of professional knowledge and personal-care philosophies [35]. Taking into account the aforementioned benefits of taking vertical positions and the barriers reported by midwives in their use in practice, we see that there is a need to improve midwifery’s leadership and management issues. It seems that the leaders should support staff in the use of modern and effective techniques and maintain high qualifications for staff and high standards for the equipment used in delivery rooms.

In a study conducted on Italian midwives working in the delivery room, it was shown that midwives’ knowledge of the purpose of individual vertical positions is quite good, and among the difficulties in applying pain-relief techniques, midwives mentioned, among others things, the relationships with other employees; the presence of a partner at birth; and certain characteristics of a woman that hinder cooperation, such as personality, behavior, physical condition [36].

Adversely to our studies, Italian midwives indicated the presence of a partner as a hindrance and obstacle, and in our research, over 70% of Polish midwives considered the role of the accompanying person to be important. In addition, 24% of them considered that establishing better cooperation with an accompanying person is a factor increasing the role of non-pharmacological pain-relief agents.

In the same Italian study, and likewise in our research, midwives indicated that they needed additional training on vertical positions because, despite having a master’s degree and specialization, they do not feel prepared to use them in working with a patient [36]. It is worth emphasizing, however, that in our research, most midwives assessed their degree of preparation for the use of vertical positions rather well.

Among the other non-pharmacological methods to relieve labor pain are warm compresses, acupuncture, music therapy, and massage [6]. Brazilian researchers have shown that patients are very interested in using non-pharmacological methods to relieve labor pain, but most of them did not know how to use them [37].

In Ethiopian research, slightly more than 59% of the surveyed birth-room employees use non-pharmacological methods to relieve labor pain [38]. Our own research has shown, however, that most midwives use the previously mentioned methods, and this is mainly massage and music therapy.

The discussed methods of relieving labor pain are important in the assessment of satisfaction of patients after natural childbirth [39]. It is worth noting that midwives should use these methods as often as possible and encourage patients to use pain-relieving techniques as an alternative to medical solutions.

The results of our research show that the frequency of vertical positions and massage of the pregnant woman’s back depends on the level of burnout of midwives. In a group of midwives who do not use vertical positions, the values indicating burnout are the highest in professional effectiveness, disappointment, and lack of commitment to customer relations. These midwives are not involved in cooperation with patients probably due to burnout. The use of vertical positions and massage requires commitment and willingness to cooperate. The literature reports that the level of burnout among midwives varies from 19.1% in Norway to 65% in Australia [40].

There was no relationship between the use of vertical positions and other non-pharmacological methods of relieving labor pain, and job satisfaction and self-efficacy. In our research, midwives’ job satisfaction is fairly high. Adversely to our studies, only 45% of midwives working in Ethiopia were satisfied with their job; 42% were satisfied with relationship with management and job requirements [41]. Therefore, it is important to take care of their job satisfaction by shaping proper professional relationships, motivating, and appreciating. Undoubtedly, leaders can play an important role here by shaping working conditions and setting their standards. Foreign studies confirmed the above thesis that the job satisfaction of midwives and the willingness to stay in a given job depend on what type of the leadership is being cultivated amongst the management [42]. In turn, Iranian research reports that a lack of job satisfaction has a significant negative impact on the quality of communication and patient care [43]. Swedish midwives, when assessing job satisfaction, emphasized the lack of professional recognition instead, drawing attention to the perceived conflict of roles and the feeling of burnout [44]. A Polish study of 350 nurses and midwives indicated that the average burnout rate was 34.7 per 100 points, and the most important factor related to burnout was emotional exhaustion. It has also been proven that job and life satisfaction affect the level of burnout [45].

The presented study has its limitations, which should be taken into account when interpreting the results. Firstly, these are cross-sectional studies, based solely on the study of self-assessment. Although the standardized research tools used in this study are sensitive instruments designed to detect different states and characteristics, all responses focus on the subjective feelings of the respondents rather than objective criteria, thus creating the risk of false-positive results.

Secondly, the study was conducted online, meaning that the researchers had no influence on the representation of the group (participation was voluntary; people interested in the topic undertook to complete the survey in the Google form). It is also important to note that the average time spent on the Internet is highest amongst young people, and this may affect the overall age representation of the sample.

When recommending the use of vertical positions and other methods of pain relief, it should be remembered that reducing the pain and anxiety felt during childbirth can contribute to establishing good relations between mother and newborn and between mother and midwife conducting the childbirth, enables women to increase control over the experience of childbirth, and encourages participation in the process of childbirth [46,47].

## 5. Conclusions

In conclusion, bearing in mind the objectives of our research, we have shown that the midwives we studied, especially those with a master’s degree, have a fairly good knowledge of vertical positions, and in their practice, they mainly use the knee–elbow position, on all fours, standing, or sitting.

Among other methods to relieve pain during natural childbirth, they most often offer a massage and music therapy.

The main factors leading to the abandonment of the use of different positions and other methods to relieve the pain are lack of cooperation with the staff and consent on the part of the management, lack of sufficient and necessary equipment in the delivery room, and lack of skills on the part of the midwife.

The respondents indicated the importance of training in the use of vertical positions and other non-pharmacological methods of pain relief.

A higher level of knowledge and preparation for using vertical positions is associated with a higher assessment of self-efficacy and job satisfaction, and also it is associated with a lower level of burnout, especially for measures of disappointment and lack of professional effectiveness.

According to this research, midwives are of the opinion that the mothers giving birth are, on average, prepared for childbirth. Of great importance is the preparation of the pregnant woman and the previous experience of the midwife, and less relevant is the presence of an accompanying person during childbirth.

This study showed the need to introduce changes in the organization of the midwife’s work. The role of midwifery’s leadership is important. Good leadership leads to midwives feeling supported and motivated; creates appropriate working conditions for them; and enables them to deepen their knowledge, professional independence, and opportunities for self-realization. Therefore, it is necessary to educate midwives and their leaders in this area. Good leadership is an important element in building the importance of midwife’s work in the delivery room.

## Figures and Tables

**Figure 1 ijerph-19-15824-f001:**
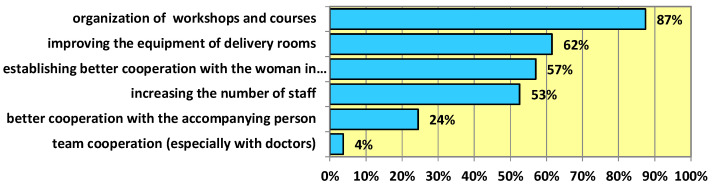
Factors increasing the role of non-pharmacological pain-relief agents.

**Table 1 ijerph-19-15824-t001:** General characteristics of the surveyed midwives.

Age (Years)	Multiplicity N = 135	Percentage %
22–30	60	44.4
31–40	27	20.0
41–50	26	19.3
Above 50	22	16.3
Sex		
Woman	134	99.3
Man	1	0.7
**Economic status**		
very good	30	22.2
quite good	101	74.8
low	4	3.0
**Education**		
post-secondary education	4	3.0
undergraduate education	50	37.0
postgraduate education	80	59.3
doctoral education	1	0.7
**Marital status**		
single	26	19.3
in a relationship, with children	78	57.8
in a relationship, no children	31	23.0
**Work experience in years**		
0–5 years	48	35.6
6–10 years	23	17.0
11–20 years	26	19.3
above 20 years	38	28.1
**Professional specialization**		
yes	70	51.9
in making	31	23.0
no	34	25.2
**GSES Questionnaire**	𝑥 ± *s*	
self-efficacy assessment	30.0 (± 3.3)	
**LBQ occupational burnout questionnaire**		
disappointment	13.4 (± 5.7)	
lack of professional effectiveness	13.1 (± 4.0)	
no commitment to relationships with clients	15.0 (± 4.7)	
psychophysical exhaustion	18.1 (± 5.9)	
assessment of job satisfaction	4.7 (± 1.0)	

**Table 2 ijerph-19-15824-t002:** Factors influencing the course of labor.

Influence of the Midwife’s Relationship with the Patient on the Course of Labor	Multiplicity, N	Percentage, %
yes	131	97.0
it does not matter	2	1.5
I have no opinion on this	2	1.5
**Difficulty in using pain-relief techniques**	**Multiplicity N**	**Percentage ^1^**
disagreement on the part of colleagues	91	67.4
disapproval of the head nurse or the chief of the department	79	58.5
lack of equipment, necessary materials, employees	65	48.1
insufficient knowledge of the midwife	61	45.2
lack of abilities	53	39.3
concerns on the part of the birthing woman	46	34.1
lack of time	45	33.3
lack of choice	1	0.7
**The level of preparation of women in labor for childbirth**	**Multiplicity N**	**Percentage**
very low	7	5.2
low	37	27.4
average	79	58.5
high	12	8.9
**Factors influencing the behavior of the woman in labor during childbirth**	**Multiplicity N**	**Percentage ^1^**
preparation for childbirth	129	95.6
previous obstetric experiences	113	83.7
presence of an accompanying person	107	79.3
involvement of medical staff	103	76.3
age of the woman in labor	45	33.3
the socioeconomic status of the patient	44	32.6
patient’s education	41	30.4
pregnancy planning	37	27.4
**The role of the accompanying person in the course of childbirth**	**Multiplicity N**	**Percentage**
low	9	6.7
average	29	21.5
big	50	37.0
Very big	47	34.8

^1^ The sum does not have to be 100% because any number of answer variants could be indicated.

**Table 3 ijerph-19-15824-t003:** Education of surveyed midwives versus the use of non-pharmacological methods of relieving labor pain and assessment of the degree of preparation for the use of vertical positions.

The Use of Vertical Positions	Education (*p* = 0.1527)		Together
	Bachelor’s Degree	Master’s Degree	
no	3 (5.6%)	2 (2.5%)	5
sometimes	14 (25.9%)	12 (14.8%)	26
yes	37 (68.5%)	67 (82.7%)	104
together	54	81	135
**the use of vertical positions used in second stage of labor**	**Education**		** *p-* ** **value**
	**Bachelor’s degree**	**Master’s degree**	
knee–elbow position	34 (63.0%)	60 (74.1%)	0.1690
on all fours position	17 (31.5%)	43 (53.1%)	0.0133 *
standing	23 (42.6%)	36 (44.4%)	0.8317
sitting	21 (38.9%)	34 (42.0%)	0.7207
lateral	7 (13.0%)	10 (12.3%)	0.9157
squatting	7 (13.0%)	8 (9.9%)	0.5762
on the chair	3 (5.6%)	0 (0.0%)	0.0319 *
**Applying warm compresses on the perineum of the woman in labor**	**Education (*p* = 0.1384)**		**together**
	**Bachelor’s degree**	**Master’s degree**	
yes	9 (16.7%)	23 (28.4%)	32
no	33 (61.1%)	36 (44.4%)	69
sometimes	12 (22.2%)	22 (27.2%)	34
Together	54	81	135
**Applying a back massage for a pregnant woman**	**Education (*p* = 0.2426)**		**together**
	**Bachelor’s degree**	**Master’s degree**	
yes	38 (70.4%)	64 (79.0%)	102
no	8 (14.8%)	5 (6.2%)	13
sometimes	8 (14.8%)	12 (14.8%)	20
together	54	81	135
**The use of aromatherapy in the delivery room**	**Education (*p* = 0.8627)**		**together**
	**Bachelor’s degree**	**Master’s degree**	
yes	9 (16.7%)	12 (14.8%)	21
no	41 (75.9%)	61 (75.3%)	102
sometimes	4 (7.4%)	8 (9.9%)	12
together	54	81	135
**Use of music therapy in the delivery room**	**Education (*p* = 0.9901)**		**together**
	**Bachelor’s degree**	**Master’s degree**	
yes	26 (48.1%)	38 (46.9%)	64
no	13 (24.1%)	20 (24.7%)	33
sometimes	15 (27.8%)	23 (28.4%)	38
together	54	81	135
**The degree of preparation for the use of vertical positions**	**Education (*p* = 0.0228 *)**		**together**
	**Bachelor’s degree**	**Master’s degree**	
1—I have no knowledge and skills	2 (3.7%)	0 (0.0%)	2
2—I don’t really feel prepared	7 (13.0%)	8 (9.9%)	15
3—I feel averagely prepared	19 (35.2%)	19 (23.5%)	38
4—I feel rather prepared	18 (33.3%)	31 (38.3%)	49
5—I am fully prepared	8 (14.8%)	23 (28.4%)	31
together	54	81	135
**The level of woman’s preparation for the childbirth**	**Education (*p* = 0.5620)**		**together**
	**Bachelor’s degree**	**Master’s degree**	
very low	1 (1.9%)	6 (7.4%)	7
low	15 (27.8%)	22 (27.2%)	37
average	33 (61.1%)	46 (56.8%)	79
High	5 (9.3%)	7 (8.6%)	12
Together	54	81	135
**The role of the accompanying person in the course of childbirth**	**Education (*p* = 0.0184 *)**		**together**
	**Bachelor’s degree**	**Master’s degree**	
very low	8 (14.8%)	1 (1.2%)	9
average	12 (22.2%)	17 (21.0%)	29
big	18 (33.3%)	32 (39.5%)	50
very big	16 (29.6%)	31 (38.3%)	47
together	54	81	135

The *p* < 0.05 level was assumed as a statistically significant relationship (*).

**Table 4 ijerph-19-15824-t004:** Difficulties in the use of pain-relief techniques according to the interviewed midwives in relation to their seniority.

Difficulties in Application of Pain-Relief Techniques	Seniority				*p*-Value
	0–5 years	6–10 years	11–20 years	≥20 years	
disagreement on the part of colleagues	38 (79.2%)	17 (73.9%)	19 (73.1%)	17 (44.7%)	0.0053 **
disapproval of the head nurse or the head of the department	38 (79.2%)	14 (60.9%)	13 (50.0%)	14 (36.8%)	0.0009 ***
lack of equipment, materials, and employees	30 (62.5%)	13 (56.5%)	8 (30.8%)	14 (36.8%)	0.0213 *
insufficient knowledge of the midwife	25 (52.1%)	10 (43.5%)	14 (53.8%)	12 (31.6%)	0.2055
lack of abilities	28 (58.3%)	9 (39.1%)	6 (23.1%)	10 (26.3%)	0.0050 **
concerns on the part of the woman giving birth	15 (31.3%)	11 (47.8%)	6 (23.1%)	14 (36.8%)	0.3035
lack of time	16 (33.3%)	8 (34.8%)	7 (26.9%)	14 (36.8%)	0.8701
**The role of the accompanying person in the course of childbirth**	**seniority (*r*_S_ = −0.19; *p* = 0.0277 *)**				**together**
	**0–5 years**	**6–10 years**	**11–20 years**	**≥20 years**	
Low	2 (4.2%)	0 (0.0%)	1 (3.8%)	6 (15.8%)	9
average	7 (14.6%)	4 (17.4%)	7 (26.9%)	11 (28.9%)	29
Big	23 (47.9%)	5 (21.7%)	11 (42.3%)	11 (28.9%)	50
very big	16 (33.3%)	14 (60.9%)	7 (26.9%)	10 (26.3%)	47
together	48	23	26	38	135
**Factors increasing the role of non-pharmacological factors in pain-relief measures**	**seniority**				***p*-value**
	**0–5 years**	**6–10 years**	**11–20 years**	**≥20 years**	
organization of workshops and courses	46 (95.8%)	22 (95.7%)	21 (80.8%)	29 (76.3%)	0.0203 *
improving the equipment of delivery rooms	37 (77.1%)	17 (73.9%)	10 (38.5%)	19 (50.0%)	0.0024 **
establishing better cooperation with the woman in labor	29 (60.4%)	11 (47.8%)	16 (61.5%)	21 (55.3%)	0.7330
increasing the number of staff	23 (47.9%)	13 (56.5%)	14 (53.8%)	21 (55.3%)	0.8759
better cooperation with the accompanying person	14 (29.2%)	6 (26.1%)	6 (23.1%)	7 (18.4%)	0.7088
team cooperation (especially with doctors)	3 (6.3%)	1 (4.3%)	0 (0.0%)	1 (2.6%)	0.5679

The *p* < 0.05 level was assumed as a statistically significant relationship (*); *p* < 0.01 is a highly significant relationship (**); *p* < 0.001 is a very highly statistically significant relationship (***).

**Table 5 ijerph-19-15824-t005:** Self-efficacy, job satisfaction, and the level of occupational burnout versus the use of vertical positions and the degree of preparation for the use of vertical positions.

Psychometric Measures (Points)	Use of Vertical Positions						*p*-Value
	No (*N* = 5)		Sometimes (*N* = 26)		Yes (*N* = 104)		
	$\bar{x}$	± SD	$\bar{x}$	± SD	$\bar{x}$	± SD	
**GSES Questionnaire**							
self-efficacy assessment	31.6	4.6	29.8	3.9	29.9	3.0	0.7706
**LBQ questionnaire**							
disappointment	18.2	8.9	16.5	5.5	12.4	5.3	0.0011 **
lack of professional effectiveness	11.2	4.6	16.1	3.5	12.4	3.8	0.0001 ***
no commitment to relationships with clients	21.8	9.7	17.0	4.4	14.1	4.0	0.0039 **
psychophysical exhaustion	17.8	11.6	19.8	6.2	17.7	5.5	0.2632
**SSP questionnaire**							
assessment of job satisfaction	4.4	1.2	4.4	1.0	4.8	1.0	0.1749
**Psychometric measures (points)**	**The degree of preparation for the use of vertical positions**						
	r_s_ (*p*-value)						
**GSES Questionnaire**							
self-efficacy assessment	0.34 (*p* = 0.0000 ***)						
**LBQ occupational burnout questionnaire**							
disappointment	−0.34 (*p* = 0.0001 ***)						
lack of professional effectiveness	−0.38 (*p* = 0.0000 ***)						
no commitment to relationships with clients	−0.26 (*p* = 0.0023 **)						
psychophysical exhaustion	−0.14 (*p* = 0.0981)						
**SSP questionnaire**							
assessment of job satisfaction—SSP	0.26 (*p* = 0.0025 **)						

*p*-test probability values calculated using the Kruskal–Wallis test; $\bar{x}$ mean, ± SD—standard deviation; r_s_—Spearman’s rank correlation coefficient. The *p* < 0.01 is a highly significant relationship (**); *p* < 0.001 is a very highly statistically significant relationship (***).

**Table 6 ijerph-19-15824-t006:** The use of non-pharmacological methods of relieving labor pain by midwives in relation to their own effectiveness, burnout, and job satisfaction.

Measure Psychometric	Use of Non-Pharmacological Methods of Pain Relief			
	Warm Compresses on the Perineum of the Woman in Labor	Back Massage of the Pregnant Woman	Aromatherapy in the Delivery Room	Music Therapy in the Delivery Room
**Self-efficacy assessment—GSES**	0.00 (*p* = 0.9915)	−0.04 (*p* = 0.6232)	0.03 (*p* = 0.6976)	−0.12 (*p* = 0.1767)
**LBQ burnout**				
disappointment	−0.02 (*p* = 0.7955)	−0.28 (*p* = 0.0012 **)	0.01 (*p* = 0.9511)	−0.06 (*p* = 0.4829)
lack of professional effectiveness	−0.08 (*p* = 0.3498)	−0.24 (*p* = 0.0050 **)	−0.03 (*p* = 0.7421)	−0.04 (*p* = 0.6334)
no commitment in relationships with customers	0.00 (*p* = 0.9648)	−0.22 (*p* = 0.0091 **)	0.04 (*p* = 0.6286)	−0.14 (*p* = 0.1145)
psychophysical exhaustion	0.00 (*p* = 0.9941)	−0.22 (*p* = 0.0115 *)	0.09 (*p* = 0.3074)	−0.01 (*p* = 0.8702)
Assessment of job satisfaction- SSP	0.12 (*p* = 0.1709)	0.15 (*p* = 0.0876)	−0.02 (*p* = 0.8550)	−0.09 (*p* = 0.3124)

The *p* < 0.05 level was assumed as a statistically significant relationship (*); *p* < 0.01 is a highly significant relationship (**).

## Data Availability

Data are available upon reasonable request.

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
