# Peer review of "The Role of Midwives in the Course of Natural Childbirth—Analysis of Sociodemographic and Psychosocial Factors—A Cross-Sectional Study"

_ijerph, 2022, doi:10.3390/ijerph192315824_

Round 1

Reviewer 1 Report

You have worked hard to carry out your research.

In the discussion, it would be good to compare/analyze each research result confirmed in this study with more abundant previous studies. 

Also, please describe more clearly the significance of nursing (nursing research, nursing education, nursing practice) of the confirmation research results.

Reviewer 2 Report

First of all, I would like to thank you for the opportunity to review this very important study.

I found the work coherent, well written and clear to the reader. I kindly ask you to review the numbering of the tables, in the current text there is no table 5, from table 4 it jumps to 6.

Reviewer 3 Report

The study entitled "The role of midwives in the course of natural childbirth analysis of sociodemographic and sychosocial factors a cross-sectional study", brings up an interesting theme, particularly I liked the approach used. The introduction presents the state of the art in an adequate way and uses significant references for the theme. The methods are well described in order to guarantee the reproducibility of the study. The results are presented clearly and objectively. The discussion is adequate and the conclusion is convergent to the manuscript's objective. I have only one consideration, I suggest that the authors create a visual abstract with the main findings of the study.

Reviewer 4 Report

This manuscript is an important contribution to the topic of non-pharmacological analgesia and positive childbirth. However, there are some inaccuracies listed below:

- line 13 (abstract): Reference is missing

- line 63: it is necessary to cite the systematic review or guideline that recommends this practice, or the recommendation should be removed.

- line 104: it is necessary to specify that this is a convenience sampling.

- lines 123-125: Are these indications for the authors? A typo? Eliminate this sentence, better introduce the results with a preamble or go directly to the subheadings.

- lines 302-303: Indicate why it is considered "interesting" or simply report this difference without expressing a latent judgment.

- lines 324-325: it is necessary to comment more on this data and provide possible explanations, in relation to what was previously reported (However, a statistically significant relationship between the level of burnout and the use of vertical positions has been demonstrated. In people who do not use vertical positions, the values indicating burnout are highest in the area of professional effectiveness (p = 0.0001), disappointment (p = 0.0001) and lack of commitment to customer relations (p = 0.004). The greater the burnout – especially for measures of disappointment and lack of professional effectiveness, the lower the self-esteem of preparation for the use of vertical positions.)

- line 350: also indicate that the population that uses the internet tends to be young.

- lines 351-354: this conclusion is not supported by the data, please remove; this study does not investigate the importance of vertical positions during childbirth, nor the reduction of pain, nor the satisfaction of mothers.

- lines 367-370: it is necessary to comment more on this data and provide possible explanations. To solve some factors (lack of cooperation with the staff and consent on the part of the management, lack of sufficient equipment in the delivery room with the necessary equipment) training is not necessary; rather, organizational leadership oriented towards health outcomes and best practices that supports the professional autonomy of midwives would be appropriate.

- lines: 372-374: this conclusion is not supported by the data.

- lines: 375-376: it is necessary to comment more on this data and provide possible explanations (as stated above).

- Discussion: there is a lack of reflections, links and possible explanations in the discussion.

- Conclusions: Conclusions need full review.

I remain at your disposal for further clarifications.

Best regards.

Author Response

Thank you for taking time to read my paper and for a fair review. We appreciate the additional comments and suggestions. We have modified manuscript according for the comments below.

Round 2

Reviewer 4 Report

The authors made the necessary corrections; however, there are still some minor inaccuracies to change listed below:

- line 458: replace "but" with ", and" to give a positive connotation to the sentence (The higher level of knowledge and preparation for using vertical positions is associated with a higher assessment of self-efficacy and job satisfaction, AND also it is associated with a lower level of burnout especially for measures of disappointment and lack of professional effectiveness).

- lines 462-463: this conclusion is not supported by the data. The authors report "Of a great importance, especially for the younger women, is the presence of an accompanying person during childbirth"; however, in lines 181-185 no reference is made to young women, so it is necessary to remove the phrase "especially for the younger women" or justify it with appropriate reference. Furthermore, "the presence of a loved one" is considered important by 79.3% of respondents, a lower percentage than other factors that are considered more relevant, such as "preparation of the pregnant woman" (95.6%) and "previous experience of the midwife "(83.7%). If the authors intend to talk about the importance of the presence of an accompanying person during childbirth, they must necessarily refer to the "preparation of the pregnant woman" and "previous experience of the midwife" considered much more relevant than "the presence of a loved one ".

I remain at your disposal for further clarifications.

Best regards.
